# A Novel Bispecific Antibody Targeting CD3 and Lewis Y with Potent Therapeutic Efficacy against Gastric Cancer

**DOI:** 10.3390/biomedicines9081059

**Published:** 2021-08-20

**Authors:** Jie Chen, Zhidi Pan, Lei Han, Yuexian Zhou, Huifang Zong, Lei Wang, Rui Sun, Hua Jiang, Yueqing Xie, Yunsheng Yuan, Mingyuan Wu, Yanling Bian, Baohong Zhang, Jianwei Zhu

**Affiliations:** 1Engineering Research Center of Cell and Therapeutic Antibody, Ministry of Education, School of Pharmacy, Shanghai Jiao Tong University, Shanghai 200240, China; jacy0214@sjtu.edu.cn (J.C.); panzhidi@sjtu.edu.cn (Z.P.); yuexianzhou@sjtu.edu.cn (Y.Z.); zhaoxiliunian@sjtu.edu.cn (H.Z.); lwangph@sjtu.edu.cn (L.W.); sun-rui@sjtu.edu.cn (R.S.); yunsheng@sjtu.edu.cn (Y.Y.); wumingyuan@sjtu.edu.cn (M.W.); weixiaobyl@sjtu.edu.cn (Y.B.); 2Jecho Biopharmaceuticals Co., Ltd., Tianjin 300450, China; lei_han@alumni.sjtu.edu.cn (L.H.); hjiang@jechoinc.com (H.J.); 3Jecho Biopharmaceutical Institute, Shanghai 200240, China; 4Jecho Laboratories, Inc., Frederick, MD 21704, USA; yxie@jechoinc.com

**Keywords:** Lewis Y, T cell-engaging bispecific antibody, m3s193 BsAb, cancer therapy

## Abstract

Lewis Y antigen, a glycan highly expressed on most epithelial cancers, was targeted for cancer treatment but lacked satisfactory results in some intractable and refractory cancers. Thus, it is highly desirable to develop an effective therapy against these cancers, hopefully based on this target. In this work, we constructed a novel T cell-engaging bispecific antibody targeting Lewis Y and CD3 (m3s193 BsAb) with the IgG-[L]-scfv format. In vitro activity of m3s193 BsAb was evaluated by affinity assay to target cells, cytotoxicity assay, cytokines releasing assay, and T cells proliferation and recruiting assays. Anti-tumor activity against gastric cancer was evaluated in vivo by subcutaneous huPBMCs/tumor cells co-grafting model and huPBMCs intravenous injecting model. In vitro, m3s193 BsAb appeared to have a high binding affinity to Lewis Y positive cells and Jurkat cells. The BsAb showed stronger activity than its parent mAb in T cell recruiting, activation, proliferation, cytokine release, and cytotoxicity. In vivo, m3s193 BsAb not only demonstrated higher therapeutic efficacy in the huPBMCs/tumor co-grafting gastric carcinoma model than the parent mAb but also eliminated tumors in the model of intravenous injection with huPBMCs. Strong anti-tumor activity of m3s193 BsAb revealed that Lewis Y could be targeted in T cell-engaging BsAb for gastric cancer therapy.

## 1. Introduction

Bispecific antibody (BsAb) has achieved a great accomplishment in fighting against diseases, including cancers. To date, three BsAbs have been on the market, and many others are in clinical research for different diseases [1]. Among them, T cell-engaging BsAbs were focused on the therapy for various cancers due to their potent activity by activating the immune system [2,3,4]. Typically, a BsAb has two arms that may target tumor association antigens on tumor cells and CD3 on T cells, which could recruit T cells into the tumor micro-environment as well as activate T cells to release cytokines, granzyme, and perforin [4].

Many cancer cell surface antigens, such as CD19, Her2, and GD2, were targeted through T cell-engaging BsAbs, showing potent anti-tumor activity [5,6,7]. Lewis Y, as a tumor-associated glycan antigen, is a type 2 blood group-related difucosylated oligosaccharide with the chemical structure [Fucα1→2Galβ1→4(Fucα1→3) GlcNAcβ1→R], and fucosylated by α1,2-fucosyltransferase at the end of glycan chain [8,9]. It was also targeted for anti-tumor treatment [10,11]. The primary expression of Lewis Y was in the embryonic period, and there was limited expression on the surface of granulocytes and epithelium in adults under the physiological condition [12]. Many studies also found that Lewis Y was in a form of glycolipid on the plasma membrane or conjugated to other cell receptors, such as EGFR, CD47, CD44, and CD147 [13,14,15,16]. Nevertheless, Lewis Y was highly expressed in the majority of carcinomas, including breast, ovary, colon, stomach, and liver cancers [17,18]. Additionally, its expression was often related to clinical stage and progression [19,20]. Several studies revealed that expression of Lewis Y on ovarian cancers could promote cancer cell proliferation by regulating expression and phosphorylation of the molecules in EGFR/PI3K/Akt signaling pathways [21,22] and as a part of integrin αv, β3, and α5β1 to enhance the adhesion and migration of ovarian cancer cells [23,24,25]. Besides ovarian cancer, 44% of cases of breast carcinomas were Lewis Y positive, and over-expression of Lewis Y was associated with a significantly decreased patient survival rate [19]. A clinical study on gastric cancer also revealed that 40–50% of cases were Lewis Y reactive and a significant strong expression on signet-ring cell carcinomas of the stomach [26].

Antibody targeting Lewis Y not only inhibited tumor growth but also enhanced anti-tumor activity when combined with taxol chemotherapy in breast cancer xenograft studies [11,27]. Anti-Lewis Y antibody also enhanced the therapeutic efficacy of celecoxib against gastric cancer by downregulation of MAPKs/COX-2 signaling pathway [28]. Researchers found that knockdown of FUT1 (gene of α1,2-fucosyltransferase), a key enzyme for Lewis Y synthesis, could down-regulate HER2 signaling via EGFR down-regulation to inhibit the proliferation of the gastric cancer cell line (NCI-N87) [29]. All these studies suggested that Lewis Y was a promising target for the therapy of epithelial cancers including ovarian, breast, and gastric cancers. Lewis Y has been evaluated as a target in several clinical trials using monoclonal antibody (mAb), antibody-drug conjugate (ADC), and chimeric antigen receptor T-Cell immunotherapy (CAR-T); however, a limited clinical benefit has been achieved so far from these trials [30,31].

T cell-engaging BsAbs have many advantages with different formats produced by various platforms [1,32]. Among them, IgG-[L]-scfv-like T cell-engaging BsAbs had potent anti-tumor activity to different tumor cell lines by targeting tumor association antigens and CD3 [6,33,34], which was confirmed by another anti-tumor BsAb in our laboratory (data not shown). Brian H. Santich et al. also reported that the IgG-[L]-scfv structure had stronger anti-tumor activity than either the IgG-like or BiTE structure for the GD2 target with appropriate inter-domain spacing and spatial configuration [7]. Thus, we were inspired by the successful reports and designed m3s193 BsAb with an IgG-[L]-scfv structure to evaluate in vitro and in vivo anti-tumor activity. Here, we report that this BsAb had an excellent curative effect against Lewis Y positive gastric cancer, showing the potential to be a new therapeutic regent in clinical applications.

## 2. Materials and Methods

### 2.1. Vectors Construction

The variable heavy chain (VH) sequences and variable light chain (VL) sequences of anti-Lewis Y antibody and Lewis Y-BsAb were derived from murine monoclonal antibody 3S193 (United States Patent 5874060) [9] and synthesized by GenScript^®^ company (Nanjing, China). VH and VL were constructed into the human IgG1, κ framework by PCR technology to obtain monoclonal antibody anti-Lewis Y, named m3s193 mAb. Furthermore, Lewis Y-BsAb was constructed by VH and VL of m3s193 into the human IgG1, κ framework with mutations of L234A, L235A, and P329G (LALA-PG) to eliminate Fc receptor and complement binding activities [35]. Anti-CD3 scfv derived from the VH and VL domains of huOKT3 [36] was fused to the C-terminal of each light chain by a linker of three G_4_S_1_ domains to form T cell-engaging BsAb, named m3s193 BsAb. An additional six G_4_S_1_ domains were added as a linker between the VH and VL domains of the scfv. For the negative control with similar IgG-[L]-scfv BsAb format, VH and VL of anti-CD22 sequence derived from M971 (United States Patent 9598492) [37] were synthesized and constructed by the same method, named M971 BsAb. All coding sequences of heavy/light chains were constructed into pcDNA3.4 vector by homologous recombination.

### 2.2. Proteins Purification

The constructed vectors were transfected into HEK293F cells using transient gene expression (TGE) technology following published protocols [38,39,40]. After 6–7 days’ culture, when the cell viability dropped to 50–60%, the supernatant was harvested and purified by MabSelect SuRe affinity chromatography. Purified mAb and BsAbs were detected by SDS-PAGE to verify the molecule weight under non-reducing and reducing conditions. All purified proteins were dialyzed overnight in phosphate-buffered saline (PBS) buffer and sterilized by filtration using a 0.22 µm filter and frozen in −80 °C freezer.

### 2.3. Binding Activity to the Target Cells

Four Lewis Y positive tumor cell lines were selected for the binding activity assay by flow cytometry. In brief, 0.2 million MCF-7, T47D, MDA-MA-231, and NCI-N87 cells were incubated with 5 μg of purified mAb and BsAb, respectively, in 4 °C for 30 min. After incubation, cells were washed twice with FACS buffer (2% FBS in PBS) and then incubated with FITC-conjugated goat anti-human IgG (H + L) secondary antibody (TheromFisherTM, Shanghai, China, Cat no. A18806) in 4 °C for 30 min. Finally, cells were washed twice with FACS buffer and analyzed with CytoFLEX cytometry (Beckman Coulter, Inc, Brea, CA, USA model number: CytoFLEX). Binding histograms were exported by CytExpert software (Beckman Coulter, Inc., Version: 2.3.0.84).

### 2.4. Binding Affinity to the Target Cells

In this assay, target cells were harvested and washed twice with FACS buffer and were equalized into round 96-well cell culture plates with 2E5 cells/well. Antibody m3s193 mAb or m3s193 BsAb was diluted 5 times, then incubated with target cells in 4 °C for 30 min under a final concentration from 500 nM to 0.0064 nM. For Jurkat cells, instead of T cells due to CD3 expression on its surface, 0.2 million cells were incubated with m3s193 BsAb diluted from 500 nM to 0.032 nM in 4 °C for 30 min. After incubation, all cells were washed and incubated with secondary antibody FITC as previously described and then detected by CytoFLEX cytometry. Finally, median fluorescence intensity was counted for each sample and analyzed by GraphPad Prism.

### 2.5. In Vitro Cytotoxicity Mediated by huPBMCs

Target cells were seeded on 96-well cell culture plates with culture medium (no phenol RPMI 1640 (Gibco, Cat no. 11835030) + 5% FBS (Gibco, Shanghai, China, Cat no. 10100147)) overnight. Purified antibodies were added with various concentrations (10-fold dilution) diluted by the culture medium (no phenol RPMI 1640 + 2% FBS). Fresh isolated huPBMCs were added with a 10:1 E/T ratio. After incubation, the supernatant was collected, and the lactate dehydrogenase activity was measured using Cytotoxic 96^®^ Non-Radioactive Cytotoxicity Assay Kit (Promega, Madison, WI, USA Cat no. G1780). All measurements were in triplicate. The percentage of cytotoxicity was calculated as follows: cytotoxicity% = (experimental lysis − spontaneous effector lysis − spontaneous target lysis)/(maximum target lysis − spontaneous target lysis) × 100, and data were analyzed by GraphPad Prism.

### 2.6. T Cells Activation with CD69 and CD25 Expression

Lewis Y positive NCI-N87 cells were seeded on 96-well cell culture plates (2E4 cells/well) overnight. On the second day, cells were incubated with fresh isolated huPBMCs (E:T = 10:1) under various concentrations of diluted antibodies and then incubated for 20 h or 90 h at 37 °C. After 20 h incubation, the cell mixture (target cells and huPBMCs) was collected and plated on round 96-well cell culture plates and then analyzed by flow cytometry using CD8-FITC, CD4-PE (Sino Biological, Beijing, China), and CD69-APC mAbs (BD Biosciences, San Jose, CA, USA) to detect the expression of CD69 on CD8^+^ and CD4^+^ T cells. In another similar assay, after 90 h incubation, the cell mixture was analyzed with the same method using CD8-FITC, CD4-PE, and CD25-APC mAbs (Sino Biological) to detect the expression of CD25 on CD8^+^ and CD4^+^ T cells. Finally, CD69^+^ and CD25^+^ percentage and median fluorescence intensity on CD8^+^ and CD4^+^ T cells were counted for each sample and analyzed by GraphPad Prism.

### 2.7. Detection of Cytokine Release

NCI-N87 cells were seeded on 96-well cell culture plates (2E4 cells/well) overnight. On the second day, cells were incubated with fresh isolated huPBMCs (E/T = 10:1) and treated with various concentrations of diluted antibodies at 37 °C. After 20 or 25 h of incubation, cells were centrifugalized with 1500 rpm (10 min), and the supernatant was harvested and frozen in −80 °C freezer immediately. Cytokine detection was performed by ELISA assay kit (R&D SYSTEM^®^, Minneapolis, MN, USA) following manual instruction. In brief, detection antibodies of IL-2 and IFN-γ were coated on ELISA high-adsorption plate with working concentration overnight. After incubation, plates were washed and blocked for 1 h. Next, IL-2, IFN-γ, and thawed supernatant were diluted, respectively, with appropriate concentrations followed by incubation for 2 h. Finally, detection antibodies of IL-2 and IFN-γ were added with working concentration for 2 h, followed by adding streptavidin-HRP, substrate solution, and stop solution as instructed by manual. The OD at 450 nm was measured with micro-plate reader (Infinite M200 Pro, Tecan, Shanghai, China), and data were analyzed by GraphPad Prism.

### 2.8. T Cell Proliferation Assay

HuPBMCs were labeled with CellTrace CFSE dye at a final concentration of 1 μM, and then CFSE-labeled huPBMCs were co-cultured with NCI-N87 cells with a 10:1 E/T ratio. This mixture was then treated with 100 ng/mL m3s193 mAb or BsAb at 37 °C. After incubation, the mixture was harvested daily from day 1 to day 4 with labeling of mouse anti-human CD3 APC-labeled (Sino Biological) mAbs and then analyzed by flow cytometry. Percentage of T cell proliferation was calculated by quantitation of the low CFSE fluorescence intensity contributed by proliferated CD3^+^ T cells.

### 2.9. Cell–Cell Association Mediated by m3s193 BsAb

Lewis Y positive NCI-N87 cells were labeled with CellTrace CFSE (Invitrogen, Shanghai, China) at a final concentration of 0.5 μM. The CD3-expressing Jurkat cells were labeled by PKH26 dye using a cell membrane labeling kit (Sigma; PKH26GL) according to the manufacturer’s protocols. Labeled cells (1 × 10^6^/mL) were mixed at equal ratio in the presence or absence of mAb or BsAb (10, 100 and 1000 ng/mL) in a 4 °C incubator for 30 min referring to previous report [32]. Population of cell–cell association was measured using flow cytometry and was quantified as the percentage of double-positive cells in the upper right quadrant of FITC-A compared with PE-A, representing the CFSE+PKH26+ cell assembly.

### 2.10. In Vivo Activity of m3s193 BsAb with huPBMCs/Tumor Cells Co-Grafting Model

Female NOD/SCID mice (6–8 weeks, Charles River Labs) were purchased and fed in accordance with guidelines from the Institutional Animal Care and Use Committee of the School of Pharmacy of Shanghai Jiao Tong University (SJTU). NCI-N87 cells (5E6 cells) were admixed with freshly isolated huPBMCs at the indicated E/T ratios of 1:3. HuPBMCs/tumor cell mixture was co-grafted (injected) s.c. (subcutaneously) in NOD/SCID mice in a total volume of 0.1 mL in RPMI medium. After co-grafting, therapy administration i.v. (intravenously) (200 μL) was performed either day 0 at the indicated doses and schedules twice a week or was continuously dosed from day 0 to day 6 every day. Tumor size was measured twice a week with a vernier caliper, and tumor volume was calculated using the approximated formula V = 0.5 × (length × width × width). Mice were euthanized when the tumor size reached 1000 mm^3^. After the mice were sacrificed, the tumor weight was detected by the electronic analytical balance and stripped tumors were photographed.

### 2.11. In Vivo Activity of m3s193 BsAb with huPBMCs Intravenous Injection Model

Female NOG mice (6–8 weeks, Charles River Labs) were injected s.c. with NCI-N87 cells (5E6 cells) and tumor volume was measured every 3 days. Mice were randomized into two groups for PBS and m3s193 BsAb (*n* = 8) when the tumor volume achieved 100 mm^3^ and injected with huPBMCs i.v. (5E6 cells). Therapy administration started 3 days after huPBMCs transfer, with i.v. injection of 100 μg m3s193 BsAb or 200 μL PBS (vehicle) every 3 days. Mice were sacrificed when GVHD happened. For huPBMCs reconstruction assay, single-cell suspension derived from mice peripheral blood after huPBMCs injected for 14 days was stained with anti-huCD45 antibody (Sino Biological), and percentage of human CD45^+^ cells was calculated and analyzed by GraphPad Prism.

### 2.12. Histologic Analysis

Tumor tissues from termination animals were fixed in 4% PFA (paraformaldehyde) overnight and embedded in paraffin. Briefly, 4μm sections were cut using a microtome (Leica) and mounted on glass slides. Samples were deparaffinized, and heat antigen retrieval was performed before immune-staining for human CD3, CD4, and CD8 using corresponding mAbs (Servicebio). The sections were counterstained with hematoxylin (Servicebio), and slides were scanned using OLYMPUS-BX53.

## 3. Results

### 3.1. Expression and Purification of the Antibodies

Bispecific antibody (m3s193 BsAb) was designed and constructed as an IgG-[L]-scfv structure, and two constructs for experimental controls were m3s193 mAb and M971 BsAb (Table 1). All three antibodies were expressed in HEK293F cells and purified by MabSelect SuRe affinity chromatography. IgG-[L]-scfv BsAb had a symmetrical structure with a normal heavy chain and a light chain with a C-terminal linker (G_4_S_1_)_3_ followed by huOKT3 scfv (Figure 1a). After expression in HEK293F cells, all proteins were purified and analyzed by SDS-PAGE to verify both quality and quantity. Two banes appeared in the antibody m3s193 mAb under the reducing condition, representing the heavy chain (about 50 kDa) and light chain (about 25 kDa) (Figure 1b). The m3s193 BsAb showed one bane at about 200 kDa under the non-reducing condition and one bane at about 50 kDa under the reducing condition (Figure 1c), which was consistent with the theoretical molecular weight of heavy and light chains of m3s193 BsAb. For the negative control M971 BsAb, two separate banes under the reducing condition were observed by SDS-PAGE analysis (Figure 1d), due to size differences of heavy and light chains. From these results, it was concluded that all three proteins were well expressed in mammalian cell line HEK293F with reasonable yield. After one-step purification, product purity could reach above 95% for BsAb that met the requirement for in vitro and in vivo assays (Table 1, Appendix A).

### 3.2. Binding Activity of Purified Proteins

Previous studies revealed that Lewis Y was expressed on breast and gastric cancers [19,26]. Three breast cancer cell lines (MCF-7, T47D, MDA-MB-231) and one gastric cancer cell line (NCI-N87) were used to detect the expression of Lewis Y and to analyze the binding activity of purified antibodies. We found that all cancer cell lines selected expressed Lewis Y and had bound m3s193 mAb and m3s193 BsAb in the flow cytometry assay (Figure 2a). On the MDA-MB-231 cell line, m3s193 BsAb had a slightly higher fluorescence intensity than the m3s193 mAb. For the negative control M971 BsAb, no binding to the NCI-N87 cell line was observed (Figure 2a). Next, we detected the binding affinity of m3s193 mAb and m3s193 BsAb to the target tumor cells and found that both antibodies had a similar binding affinity curve to Lewis Y positive tumor cell lines, suggesting that BsAb maintained the high affinity of parent mAb (Figure 2b). The result of binding to the Jurkat cell line also revealed that m3s193 BsAb had a high binding affinity to Jurkat cells with a low EC_50_ value (1.035 nM) (Figure 2b), indicating that the bivalent binding model of the IgG-[L]-scfv structure possibly maintained the high affinity of BsAb to target cells.

### 3.3. T Cell Cytotoxicity Mediated by m3s193 BsAbs

We designed the bispecific antibody as an IgG-[L]-scfv structure and observed potent T cell redirecting cytotoxicity to target tumor cells, as previously reported [7]. To MCF-7 and T47D cells, potent cytotoxicity was observed with m3s193 mAb, potentially due to their sensitivity to ADCC activity by m3s193 mAb. M3s193 BsAb achieved similar cytotoxicity as m3s193 mAb with a comparable EC_50_ value and a slightly higher potency (Table 2, Figure 2c). To MDA-MB-231 and NCI-N87 cells, in contrast, m3s193 BsAb had much better T cell-mediated cytotoxicity than m3s193 mAb, with a lower cytotoxicity EC_50_ value and a remarkably higher cytotoxicity in cell killing percentage. These results demonstrated that m3s193 BsAb had better in vitro activity than m3s193 mAb. In an independent cytotoxicity comparison study between the two BsAbs, no cytotoxicity was observed with M971 BsAb except basal noise, while m3s193 BsAb reproducibly exhibited low cell killing EC_50_ at 2.673 ng/mL. This suggested that T cell-mediated tumor cell killing by the IgG-[L]-scfv was dependent on the target expression on tumor cells. M3s193 BsAb would be expected to have less non-specific toxicity and minimum cytotoxicity to Lewis Y negative expression tissues in the body. The slight EC_50_ differences (9.315 and 2.673 ng/mL) in two experiments observed in NCI-N87 cytotoxicity mediated by m3s193 BsAb might be due to the differences of huPBMCs donors.

### 3.4. T Cell Activation Mediated by m3s193 BsAb

T cell-engaging BsAbs could activate T cells with the expression of CD69 and CD25 as previously reported [41]. By flow cytometry assay, CD69 expression on T cells was activated by m3s193 BsAb but not by m3s193 mAb. The percentage of CD8^+^CD69^+^ in CD8^+^ and CD4^+^CD69^+^ in CD4^+^ achieved nearly 100% (Figure 3a). MFI data further proved the potent activation of T cells by m3s193 BsAb with high CD69 expression at the concentration from 100 to 10,000 ng/mL, while no CD69 expression on CD8^+^ and CD4^+^ T cells was detected for the control mAb even at a high concentration. This observation was further confirmed by the negative control M971 BsAb (Figure 3b). This suggested that m3s193 BsAb but not m3s193 mAb could activate T cells, and the activation was dependent on antigen expression on the tumor cells. By the detection of CD25, a later activation marker on T cells, we found that a similar result was observed as CD69 expression on CD8^+^ and CD4^+^ T cells mediated by m3s193 BsAb and M971 BsAb (Figure 3c), demonstrating the continuous activation of CD8^+^ and CD4^+^ T cells by m3s193 BsAb from the early to later phase but not the control mAb and M971 BsAb. To eliminate the impact from huPBMCs’ auto-activation, a quantity of CD69 positive T cells in CD8^+^ and CD4^+^ cells was detected with the mixture of huPBMCs and m3s193 BsAb. The result showed that only less than 0.4% CD69-positive T cells in CD8^+^ and CD4^+^ T cells was detected even at a concentration of 1000 ng/mL of m3s193 BsAb (Appendix Aa,b). Both CD8^+^ and CD4^+^ T cells play critical roles in preventing the body from pathogens in the immune system. Thus, m3s193 BsAb rather than m3s193 mAb and M971 BsAb could activate the T cells to lead potent in vitro cytotoxicity.

### 3.5. Cytokine Release, Proliferation, and Recruiting Mediated by m3s193 BsAb

Cancer immunotherapy by cytokines has been confirmed by a number of reports [42,43]. To evaluate if cytokines were released upon T cell activation by m3s193 BsAb, IL-2 and IFN-γ detection assays were performed after T cell activation. Compared with control mAb and M971 BsAb, m3s193 BsAb could induce IL-2 and IFN-γ release but not controls (Figure 4a,b). This result also proved that m3s193 BsAb had a strong anti-tumor activity by the release of IL-2 and IFN-γ from huPBMCs to proliferate T cells and kill tumor cells. In contrast, there was no non-specific cytokine release and cell killing to Lewis Y negative expression cells (Figure 2c and Figure 4b).

We also found that m3s193 BsAb could induce CD3^+^ T cell proliferation from 10% to 40% after 2 to 4 days of incubation, while the control mAb only reached less than 5% proliferation percentage even at the fourth day (Figure 4c). This might be due to the proliferation signaling to T cells mediated by IL-2 which were released by T cell-engaging BsAb activation [44].

T cell-engaging BsAb could recruit T cells into the tumor micro-environment to form the immune synapse and then activate T cells to release cytokines, granzyme, and perforin to kill tumor cells [45]. Thus, we used Jurkat cells instead of T cells due to CD3 expression on its surface to evaluate the recruiting ability of m3s193 BsAb. We found that m3s193 BsAb could mediate strong cell–cell association from the concentration of 10 to 1000 ng/mL, while m3s193 mAb appeared similar to the group of “No addition” (Figure 4d). This result indicated that m3s193 BsAb rather than m3s193 mAb could recruit T cells to tumor cells in the in vitro activity assessment, which provided the potential of T cell infiltration into the tumor micro-environment in vivo.

### 3.6. In Vivo Anti-Tumor Activity

#### 3.6.1. Tumor Inhibitory Activity of m3s193 BsAb

To evaluate anti-tumor activity in vivo, the gastric cancer cell line NCI-N87 was inoculated in the huPBMCs/tumor co-grafting model with two drug delivery procedures in the assessment of m3s193 BsAb. From this initial study, we found that m3s193 BsAb had excellent anti-tumor activity compared with control group PBS in both drug delivery procedures (Figure 5a,e). In the first, twice a week delivery, the body weight of mice had no fluctuation between mice in both the testing group and control group (Figure 5b). In the second, continuous delivery, body weight fluctuations both in PBS and m3s193 BsAb groups were observed (Figure 5f). The differences in the changes of weight in mice due to continuous drug delivery suggested interval drug delivery for the following in vivo activity evaluation. Tumor size and weight in the m3s193 BsAb group also proved potent anti-tumor activity in both delivery procedures (Figure 5c,d,g,h). All these data suggested that m3s193 BsAb had strong anti-tumor activity for eliminating tumors with no obvious toxicity in mice.

#### 3.6.2. Remarkable Tumor Inhibitory Activity of m3s193 BsAb

In vivo anti-tumor activity was further evaluated with various dosages of m3s193 BsAb compared with m3s193 mAb. BsAb m3s193 could significantly inhibit tumor growth with dose dependency (Figure 6a), while m3s193 mAb did not inhibit tumor growth at 20 μg dosage close to the “PBS group” (*p* > 0.05) (Figure 6b). Mean tumor growth curve and tumor size image also revealed potent anti-tumor activity of m3s193 BsAb under various dosages with significant *p* value (Figure 6b,c). There were no body weight differences among five groups, suggesting that m3s193 BsAb had a minimum toxicity (Figure 6d). On the other hand, tumor weight reducing in the BsAb group showed better inhibitory tumor activity of m3s193 BsAb than the parent mAb (Figure 6e). These data illustrated that systematic administration of m3s193 BsAb could inhibit the Lewis Y-expressing NCI-N87 xenograft tumor growth compared with the parent mAb. By immune-histochemical assay, there was no existence of CD8^+^ and CD4^+^ T cells in the tumor micro-environment in the PBS and mAb (20 μg) groups, but there was existence in the BsAb (20 μg) group (Figure 6f), which suggested that T cells, particularly for CD8^+^ and CD4^+^ T cells, could survive for a long time when activated by m3s193 BsAb. This could be one of the reasons for inhibiting tumor growth in the m3s193 BsAb group.

#### 3.6.3. M3s193 BsAb Had Potent Anti-Tumor Activity in Therapeutic Study

In the tumor/huPBMCs co-grafting animal model, m3s193 BsAb could inhibit tumor growth and had much stronger anti-tumor activity than m3s193 mAb. In this model, huPBMCs were mixed with tumor cells and were forcedly infiltrated by human factor. Under normal physiological conditions, in the tumor micro-environment where some tumors, especially non-immunoreactive tumors, were, T cells only had a very low level or almost did not exist [46,47]. To mimic this condition, we performed the therapeutic study by only subcutaneously implanting tumor cells into mice before huPBMCs and therapy administration were injected. First, we detected the reconstruction of huPBMCs in NOG mice and found that all mice (*n* = 16) were successfully reconstructed with about 20–60% of huCD45 cells in mice peripheral blood after fourteen days of huPBMCs transfer (Figure 7a), which ensured that all mice could be used for the effectiveness evaluation of m3s193 BsAb. From the tumor growth curve, m3s193 BsAb could eliminate tumors even though the tumor volume reached about 600 mm^3^ (Figure 7b,c). All tumors in the m3s193 BsAb group were much smaller when tumors were stripped after study termination (Figure 7d). The mean weight of stripped tumors in the m3s193 BsAb group further confirmed this result (Figure 7e). These data suggested that m3s193 BsAb could eliminate tumors in vivo by huPBMCs reconstruction in mice. Immune-histochemical results revealed that CD3^+^ T cells existed in the PBS group and were mainly around tumor stroma (red oval circle) (Figure 7f). In the m3s193 BsAb group, there was only a small residual tumor tissue (yellow oval circle) surrounded by T cells, which illustrated that m3s193 BsAb could mediate the infiltration of T cells derived from peripheral blood into the tumor micro-environment. This result could explain the reason for tumor elimination in the m3s193 BaAb group by mediating T cell infiltration.

## 4. Discussion

Bispecific antibody (BsAb) plays an important role in immunotherapy for cancer and other diseases. In this report, we constructed a novel T cell-engaging BsAb targeting Lewis Y and CD3 antigen to evaluate anti-tumor activity for the first time. The BsAb (m3s193 BsAb) had similar binding affinity to Lewis Y positive cancer cell lines compared to its parent mAb, due to the IgG-[L]-scfv structure keeping two Fab binding domains of normal antibody. Experimental results revealed that m3s193 BsAb could activate CD69 and CD25 expression on T cells and induced cytokine release to mediate cytotoxicity in vitro. Although Lewis Y was expressed on granulocytes in peripheral blood, m3s193 BsAb could not activate CD8^+^ and CD4^+^ T cells with CD69 expression when only co-incubated with huPBMCs in the absence of target cells (Appendix A a,b). These data further indicated the specificity of IgG-[L]-scfv-derived T cell-engaging BsAbs with two binding scfvs of anti-CD3. Besides activation, m3s193 BsAb, rather than the control mAb, could mediate cell–cell association, which validated the basic function of T cell-engaging BsAbs, as reported previously [48]. By shortening distance between T cells and tumor cells, T cell-engaging BsAbs could mediate the formation of an immune synapse to activate T cells [49]. Thus, a closer distance is mandatory for a proper induction of T cell activation [50].

In both drug delivery procedures in the animal in vivo study, m3s193 BsAb could significantly inhibit tumor growth with no body weight fluctuation in a twice-a-week delivery procedure, suggesting that interval therapy was suitable in the following in vivo experiment (Figure 5). In following the tumor/huPBMCs co-grafting model, m3s193 BsAb performed a similar tumor inhibitory activity with dose dependency, while the control mAb did not show activity at the high dosage, which was due to the existence of CD8^+^ and CD4^+^ T cells in the tumor micro-environment mediated by BsAb (Figure 6). M3s193 BsAb also eliminated large size tumors (600 mm^3^) in mice by inducing CD3^+^ T cell infiltration, as previously reported in a similar solid tumor treatment by carcinoembryoin antigen T cell-engaging BsAb [51,52].

Targeting tumor-association glycan would be a potential therapy strategy due to its fundamental and crucial physiological functions in the genesis and development of cancers [53]. In addition to Lewis Y, several therapies targeting glycan neo-antigens have been in clinical trials, such as monosialodihexosylganglioside (GM3 containing Neu5Gc), gangliosides (GD3 and GD2), and Tn glycoform of MUC1 [54,55,56,57]. Thus, this work not only provided a new therapy targeting Lewis Y to make up for the deficiency of mAb therapy but also illustrated that targeting tumor glycan-antigen would contribute to a new era of cancer-targeted therapy where BsAbs may play a potentially crucial role.

## 5. Conclusions

Our findings provide more meaningful evidence for the treatment of gastric cancer by targeting a tumor-associated glycan-antigen. All these data support the idea that the Lewis Y target could be developed into a preclinical anti-tumor study by redirecting T cells to activate the immune response.

## Figures and Tables

**Figure 1 biomedicines-09-01059-f001:**
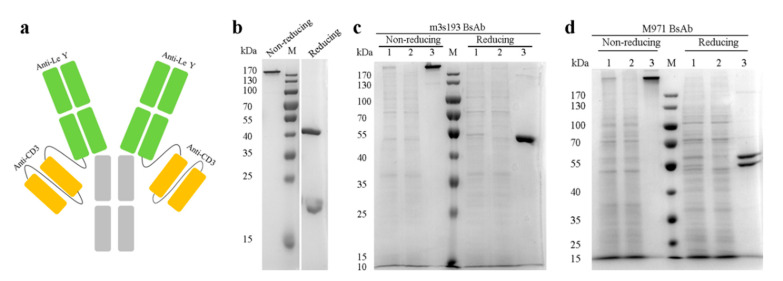
Structure schematic of m3s193 BsAb and purification of the proteins. (**a**) Molecular design of m3s193 BsAb with IgG-[L]-scfv structure. (**b**) SDS-PAGE analysis of purified m3s193 mAb. (**c**) SDS-PAGE analysis of purified m3s193 BsAb. (**d**) SDS-PAGE analysis of purified M971 BsAb. Lane 1, the supernatant before purification; lane 2, the flow through after purification; and lane 3, the elution of proteins.

**Figure 2 biomedicines-09-01059-f002:**
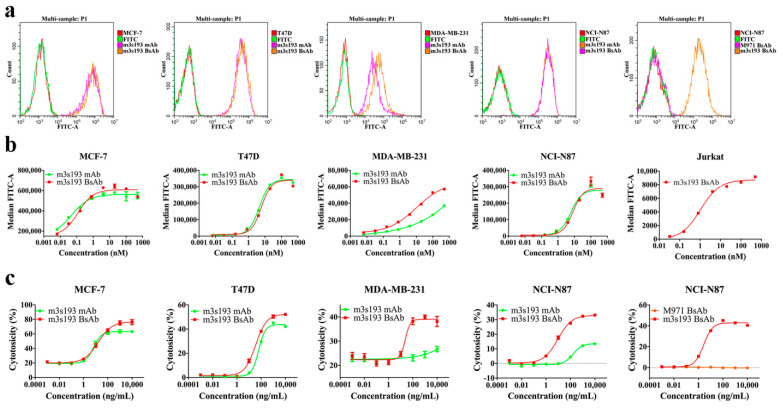
Flow cytometry binding analysis and in vitro cytotoxicity mediated by huPBMCs. (**a**) FACS histograms of Lewis Y expression in different cancer cell lines and binding of m3s193 mAb, m3s193 BsAb, and M971 BsAb to the cells. (**b**) Binding affinity curve of m3s193 mAb and m3s193 BsAb to four target tumor cell lines and Jurkat cell line. Median fluorescence intensity (MFI) of each sample indicated the binding potency was analyzed by GraphPad Prism. (**c**) Cytotoxicity to four target tumor cells mediated by huPBMCs in the presence of proteins. Data are shown as means ± SD with three replicates.

**Figure 3 biomedicines-09-01059-f003:**
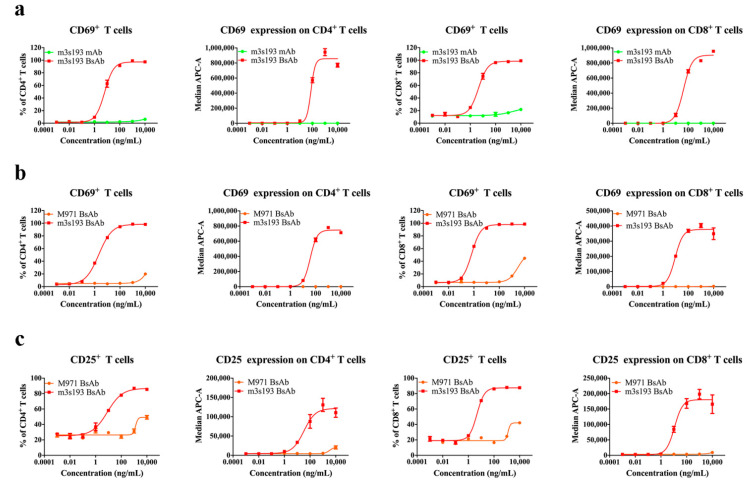
Activation of CD69 and CD25 expression on CD8^+^ and CD4^+^ T cells. The expression frequency and intensity of CD69 or CD25 on CD8^+^ and CD4^+^ T cells after 20 h (**a**,**b**) or 90 h (**c**) of co-incubation of NCI-N87 cells and huPBMCs. Data are shown as means ± SD with three duplicates.

**Figure 4 biomedicines-09-01059-f004:**
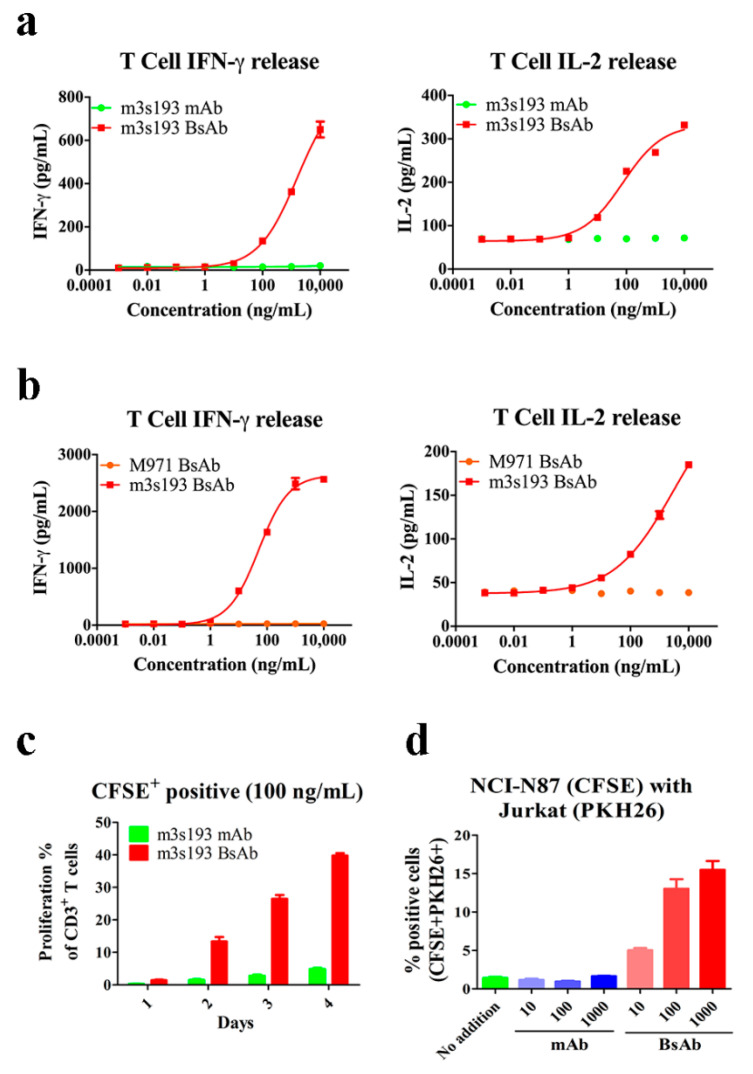
Cytokines release, proliferation, and recruiting mediated by m3s193 BsAb. (**a**) Cytokines release after 20 h incubation. (**b**) Cytokines release after 25 h incubation. (**c**) Proliferation of CD3^+^ T cells with 100 ng/mL of m3s193 BsAb or mAb. (**d**) Cell–cell association mediated by m3s193 BsAb or mAb. Data are shown as means ± SD with three duplicates.

**Figure 5 biomedicines-09-01059-f005:**
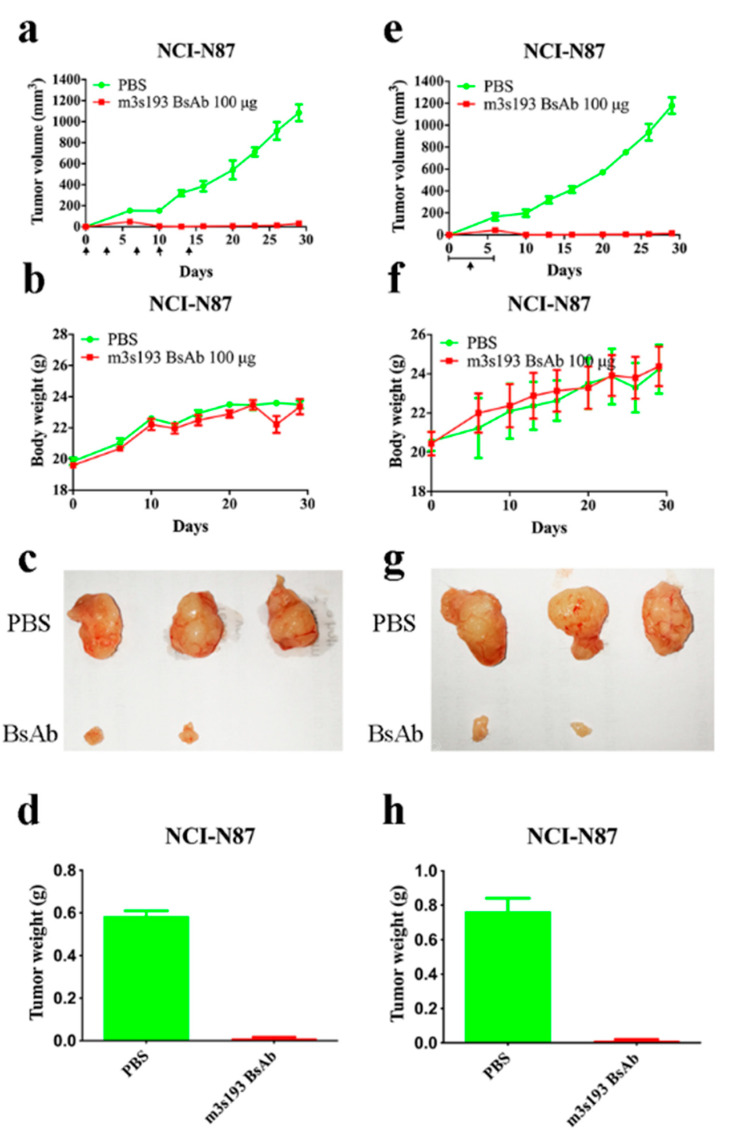
In vivo anti-tumor activity of m3s193 BsAb. Mice were treated i.v. with PBS (*n* = 3) and 100 μg dosage of m3s193 BsAb (*n* = 5). (**a**–**d**) Mice were treated on day 0, twice a week, 5 times. (**e**–**h**) Mice were treated on day 0–6, 7 times. (**a**,**e**) Tumor growth curve plotted according to tumor volume and time. Black arrows indicate the time of therapy. (**b**,**f**) Mice body weight variation with time course. (**c**,**g**) The digital image of stripped tumors at study termination. (**d**,**h**) Tumor weight in two groups. Data are presented as mean ± SD.

**Figure 6 biomedicines-09-01059-f006:**
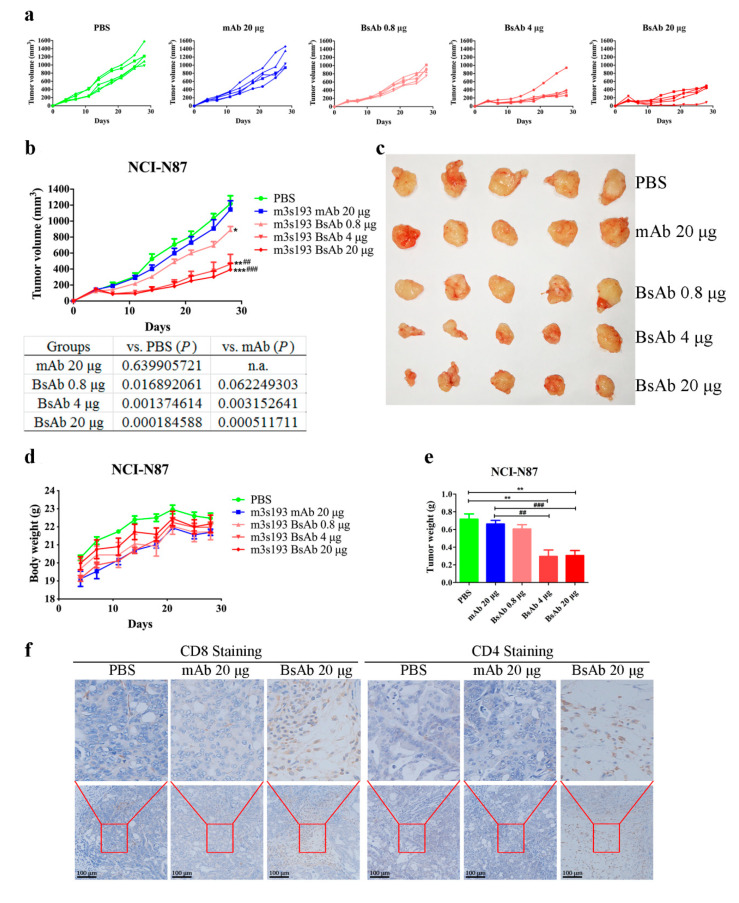
In vivo anti-tumor activity of m3s193 mAb and m3s193 BsAb. Mice were treated i.v. with PBS, mAb (20 μg), and BsAb (0.8, 4, 20 μg) on day 0, twice a week, 6 times. (**a**) Individual tumor growth curve with time course. Data are presented as measured tumor volume from each mouse in five groups. (**b**) Mean tumor growth curve with time course. Data are presented as mean ± SD. Statistical significance of experiment groups compared to PBS and mAb group was analyzed by Student’s t-test with two-tailed double sample isovariance hypothesis in Microsoft Excel, and *p* < 0.05 was considered statistically significant. (**c**) The digital image of stripped tumors. (**d**) Mice body weight of five groups with time course and presented as mean ± SD. (**e**) Weight of stripped tumors in five groups and presented as mean ± SD. Differences between samples were also tested for statistical significance by the same Student’s t-test. (**f**) Representative images of histologic analysis of explanted tumors stained for human CD8 and CD4. In this figure, *, **, and *** mean compared with the PBS group, * *p* < 0.05, ** *p* < 0.01, and *** *p* < 0.001. ##, and ### mean compared with the mAb group, ## *p* < 0.01, and ### *p* < 0.001.

**Figure 7 biomedicines-09-01059-f007:**
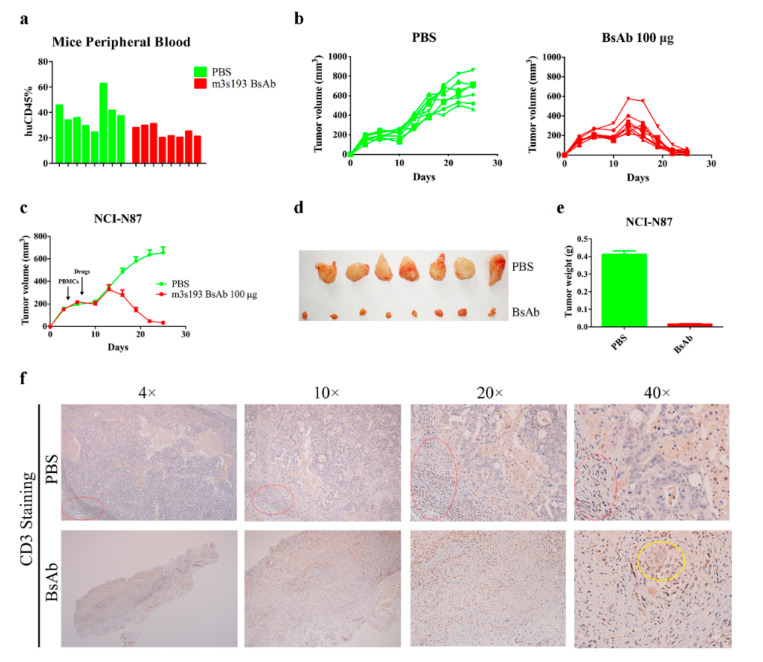
In vivo anti-tumor activity of m3s193 BsAb with i.v. transfer of huPBMCs. (**a**) Flow-cytometry analysis of huCD45^+^ cells in mice peripheral blood. (**b**) Individual tumor growth curve with time course. Data are presented as measured tumor volume from each mouse in two groups. (**c**) Mean tumor growth curve with time course. Data are presented as mean ± SD. (**d**) The digital image of stripped tumors. (**e**) Weight of stripped tumors and presented as mean ± SD. (**f**) Representative images of histologic analysis of explanted tumors stained for human CD3. Red oval circle represents infiltrated huPBMCs around tumor stroma in PBS group, and yellow oval circle represents residual tumor tissue surrounded by infiltrated huPBMCs in BsAb group.

**Table 1 biomedicines-09-01059-t001:** Antibodies used in this study.

Name	Structure	Fc Mutation	Target	SEC-HPLC Purity (%)
m3s193 mAb	IgG1	No	Lewis Y	92.63%
m3s193 BsAb	IgG-[L]-scfv	LALA-PG	Lewis Y and CD3	98.44%
M971 BsAb	IgG-[L]-scfv	LALA-PG	CD22 and CD3	98.79%

**Table 2 biomedicines-09-01059-t002:** Cytotoxicity EC_50_ (ng/mL) mediated by the antibodies.

	Cells	MCF-7	T47D	MDA-MB-231	NCI-N87	NCI-N87
Proteins						
m3s193 mAb		6.689	62.52	2.296E + 15	162.5	
m3s193 BsAb		162.5	35.89	23.98	9.315	2.673
M971 BsAb						ND*

ND*: No cytotoxicity was observed.

## Data Availability

The dataset supporting the conclusions of this article is included within the article.

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
