# Peer review of "A Novel Bispecific Antibody Targeting CD3 and Lewis Y with Potent Therapeutic Efficacy against Gastric Cancer"

_biomedicines, 2021, doi:10.3390/biomedicines9081059_

Round 1
Reviewer 1 Report
The manuscript titled “A novel bispecific antibody targeting CD3 and Lewis Y with potent therapeutic efficacy against gastric cancer” describes constructing a novel T cell engaging bispecific antibody targeting Lewis Y and CD3 (m3s193 BsAb) with the IgG-[L]-scfv format. Anti-tumor activity against gastric cancer was evaluated in vivo by subcutaneous huPBMCs/tumor cells co-grafting and huPBMCs intravenous injecting models. Strong anti-tumor activity of m3s193 BsAb revealed that Lewis Y could be targeted in T cell-engaging BsAb for gastric cancer therapy. This is an important field of study based on the overall burden of the disease. The manuscript has been written very well. However, the manuscript and some figures do not have information about how the statistical analysis was done. I would recommend accepting the manuscript after major revisions. Comments: Please provide references after the following statement in the Introduction section: Among them, T cell-engaging 31 BsAbs were focused on the therapy for various cancers due to its potent activity by acti- vating immune system. In the materials and methods section include information about from where the CytoFLEX cytometry. And CytExpert software were purchased. In 2.5 (Materials and Methods), please include information about where all the reagents were purchased. Figure 2: Please increase the font size as it is difficult to interpret the results in the current form. Please provide information about how the statistical analysis was done (Separate section). Additionally, include information of statistical analysis under each Figure.Author Response
Please see the attachment.

Reviewer 2 Report
The authors constructed a novel T cell-engaging bispecific antibody targeting Lewis Y and CD3 and showed in vitro and in vivo activity of the antibody.
Results are clearly described and the efficacy of bispecific antibody in contrasting tumor growth is well described.
In my opinion the manuscript is suitable for publication in the present form.
Minor:
-if the "+" is used for cluster of differentiaton (CD) is not necessary to write positive after it (es. change "CD3+ positive T cells" to £ CD3+ T cells").
-describe the abbreviation used s.c. and i.v.
-uniform in some of the figure legends the abbreviation for intravenous administration (change "IV" to "i.v.")
-page 5-line 4: is not GVHD instead of GvDH?
